# The Bone Bridge Technique Utilizing Bone from the Lateral Wall of the Maxillary Sinus for Ridge Augmentation: Case Reports of a 1–7 Year Follow-Up

**DOI:** 10.3390/medicina59091626

**Published:** 2023-09-08

**Authors:** Won-Bae Park, Ji-Young Han, Philip Kang

**Affiliations:** 1Department of Periodontology, School of Dentistry, Kyung Hee University, Seoul 02447, Republic of Korea; wbpdds@naver.com; 2Private Practice in Periodontics and Implant Dentistry, Seoul 02771, Republic of Korea; 3Department of Periodontology, Division of Dentistry, College of Medicine, Hanyang University, 222-1 Wangsimni-ro, Seongdong-gu, Seoul 04763, Republic of Korea; 4Division of Periodontics, Section of Oral, Diagnostic and Rehabilitation Sciences Columbia University College of Dental Medicine, #PH7E-110, 630 W. 168 St., New York, NY 10032, USA

**Keywords:** bone graft, dental implant, extraction socket, lateral sinus window, ridge augmentation

## Abstract

The post-extraction socket of a periodontally compromised tooth/implant is oftentimes accompanied by a very wide-deep alveolar ridge defect. The commonly utilized treatment is ridge preservation followed by delayed implant placement 4 to 6 months after extraction. In the four cases presented in this study, a novel technique of utilizing a bone block obtained from the lateral wall of the maxillary sinus is introduced. Due to the severe localized vertical ridge deficiency, an intraoral autogenous bone block was obtained from the ipsilateral sinus bony window. After the obtained bone block was properly trimmed, it was fixed in the form of a bridge over the vertical defect by the press-fit method. In two cases, the gap between the autogenous bone and defect was filled with a particulate synthetic bone graft, and in another two cases, the gap was left without grafting. All cases were covered with a resorbable collagen membrane. At the time of re-entry after 5 to 6 months, the bone bridge was well incorporated beside the adjacent native bone and helped by the implant placement. Uncovering was performed after 3 to 6 months, and prostheses were delivered after 2 months. Oral function was maintained without any change in the marginal bone level even after the 1- to 7-year post-prosthesis delivery. This case series showed that the bone bridge technique performed using an ipsilateral sinus bony window for a localized vertical deficiency of a post-extraction socket can be used for successful vertical ridge augmentation (VRA).

## 1. Introduction

The extraction timing of the hopeless tooth affects the maintenance of residual bone height and preservation of the post-extraction socket [1]. It is common for patients to think that maintaining their teeth for as long as possible is a good option, and surgeons recommend early tooth extraction as a strategic concept if necessary to preserve the proper alveolar ridge. If the proper timing of tooth extraction is missed due to patients’ unwillingness or unfavorable circumstances, the hopeless tooth eventually becomes a periodontally compromised tooth. Periodontally compromised teeth have accompanying tooth mobility, have accompanying root exposure, are surrounded by thick inflammatory granulation tissue, and the supporting apparatus can be lost [2,3]. The extraction socket is accompanied by severe alveolar bone resorption, especially in vertical ridge dimensions [3,4]. For this treatment, socket preservation or augmentation and horizontal/vertical bone augmentation are performed. The implant is then placed in a delayed manner, and the treatment duration is greatly extended [1,5].

Guided bone regeneration (GBR) techniques using bone block and/or particulate bone with barrier membranes are commonly used for ridge augmentation in compromised post-extraction sockets with a localized vertical ridge deficiency [6,7]. Among several options, autogenous bone grafts are considered the gold standard for alveolar ridge augmentation [8]. Bone block grafts are obtained from the symphysis and ascending ramus; using tenting screws and polytetrafluorethylene (PTFE) membranes, distraction osteogenesis, and the bone ring technique have been reported as methods to improve existing vertical ridge deficiencies [9,10,11]. However, due to morbidities such as paresthesia, hypoesthesia, and bleeding [12], an intraoral donor site with less patient morbidity and complications was investigated and presented in this case report.

Recently, Park et al. proposed that the bone obtained from the lateral wall of the maxillary sinus can be used as an intraoral donor for an autogenous bone graft [13]. In addition, they showed that excellent guided bone regeneration can be achieved when used together with a resorbable collagen membrane, and they suggested that it is one of the ways to improve various peri-implant defects, especially in situations with a vertical ridge deficiency.

To the knowledge of the authors, there has not been any case report applying the bone bridge technique using the lateral sinus bony window for vertical ridge augmentation (VRA) in a compromised post-extraction socket. The purpose of this case report is to introduce cases of successful VRA using the bone bridge technique for post-extraction sockets with severe localized vertical ridge deficiencies.

## 2. Case Presentation

### 2.1. Case 1

The patient was a 38-year-old female non-smoker without systemic disease. A severe vertical bone defect was observed in the extraction socket of the right mandibular first molar using a panoramic radiograph taken before surgery. Tooth extraction was performed 3 months prior due to periodontal disease (Figure 1a). A vertical ridge deficiency of the extraction socket was observed on a panoramic image of CBCT taken before surgery. A periapical and distal lesion was observed in the #47 tooth. As such, the ascending ramus to the donor site of the autogenous bone was excluded (Figure 1b). On the images of the CBCT taken before surgery, the right maxillary sinus was clear without any existing pathology (Figure 1c), and the lateral sinus wall was maintained at an appropriate thickness. (Figure 1d).

In a clinical picture taken 3 months after #46 tooth extraction, a vertical deficiency was observed (Figure 2a). Under local anesthesia, the mucoperiosteal flap was reflected. An unhealed bone defect was observed in the extraction socket, but the proximal bone level of the adjacent teeth was well maintained (Figure 2b). A decision was made to use block-type autogenous bone as a bone graft substitute, and the lateral sinus window on the ipsilateral side was selected as the donor site. A lateral sinus window larger than the size of the bone defect at site #46 was formed (Figure 2c). The lateral sinus window was removed without perforation of the maxillary sinus mucosa (Figure 2d). In this area, the flap was closed. The harvested lateral bone lid was 2.0 × 1.5 mm in size (Figure 2e). The sharp edge of the graft was trimmed. The lateral bone lid was fixed to the extraction socket site by the press-fit method (Figure 2f). The gap between the bone lid and the extraction socket was filled with particulate bone graft substitutes (Genoss, Suwon, Republic of Korea) (Figure 2g). The grafted site was covered with a resorbable collagen membrane (Genoss, Suwon, Republic of Korea) (Figure 2h). The inner surface of the buccal flap was appropriately dissected with a #15 Bard-Parker blade, followed by a flap extension. The flap was then closed using 4-0 nylon (Figure 2i). The wound edge was not exposed, and healing was uneventful.

Clinical findings after 6 months of bone graft healing showed that the collapsed ridge was remarkably vertically restored (Figure 3a). After local anesthesia, the mucoperiosteal flap was reflected. The remaining collagen membrane was removed, and the implanted bone bridge was exposed. The bone bridge was well integrated with the adjacent native bone, and surface resorption was not observed (Figure 3b). A core biopsy was performed with a trephine drill with a diameter of 3.0 mm at the area where the implant was to be placed. The bone bridge and the lower particulate bone grafts were well fused with each other, and the dense bone quality was maintained (Figure 3c). The biopsy specimen was fixed in 10% formalin for a histopathological examination. After additional osteotomy, a Ø 4.3 × 10 mm SLA (sandblast, large grit, acid-etched)-textured implant (Implantium, Dentium, Suwon, Republic of Korea) was placed subcrestally at about 1.0 mm. The initial stability of the placed implant was achieved (Figure 3d). The flap was closed with a 4-0 catgut (Figure 3e). Uncovering was performed 3 months after the implant placement, and the prosthesis was delivered 2 months later (Figure 3f).

A micro-CT (Skyscan 1173, Kontich, Belgium, tube voltage: 130 KV, resolution: 14.91 µm, intensity: 60 µA) analysis was performed on the obtained specimen. In the micro-CT image of the specimen, the upper part is the bone bridge, and the lower part is where the bone defect used to be. The upper bone bridge maintained its original thickness with almost no surface resorption, and new bone was found to be concentrated directly below the bone bridge (Figure 4a,b). The bone graft substitute filled in the bone defect area was maintained in its original shape without deformation, and the formation of new bone was observed between the bone graft particles. Significant bone formation was also observed in the lower part of the specimen because it was adjacent to the native bone (Figure 4c,d).

The specimen was longitudinally sectioned to a thickness of 5 µm for histopathological analysis after decalcification, followed by a hematoxylin and eosin stain (H-E stain) and Masson’s trichrome stain (MT stain). Specimens were scanned by a digital scanner (Panoramic 250 Flash III; 3 DHISTECH, Budapest, Hungary) and examined using computer software (CaseViewer ver. 2.3; 3 DHISTECH). In the histopathological examination, the upper part was the bone bridge, and the lower part was the native bone of the post-extraction socket. The middle part was filled with particulate bone. It was found that bone formation started close to the bone bridge and native bone and moved to the particulate bone in the middle. However, the osteoconductive effect of the particulate bone was sufficient (H-E stain) (Figure 5a). MT staining showed that the new bone under the bone bridge was still immature bone (Figure 5b). High-magnification findings of the bone bridge showed a capillary, osteoblast, and osteocyte. Bone bridge-linked new bone was observed (MT stain) (Figure 5c). New bone formation was evident around the particulate bone, and loose connective tissue was also observed (H-E stain) (Figure 5d). Overall, histological findings were consistent with micro-CT findings.

CBCT was taken appropriately for each treatment period. In the sagittal image of the CBCT taken before surgery, the extraction socket of the #46 tooth had a very severe bone defect. A radiolucent image was observed in the distal area of the #47 tooth (Figure 6a). A vertical ridge defect was observed in the cross-sectional image of the CBCT scanned from the extraction socket of the #46 tooth (Figure 6b). Bone bridge and bone graft particles were observed in the sagittal image of CBCT taken immediately after surgery (Figure 6c). Bone bridge and particulate bone graft substitutes were observed in cross-sectional images of CBCT taken immediately after surgery (Figure 6d). In the sagittal and coronal images of CBCT taken 6 months after implant placement, the space between the bone bridge and the lower bone graft substitute was well ossified (Figure 6e,f). The implant was placed 6 months after the bone bridge technique was performed. Uncovering was performed 3 months after implant placement, and the prosthesis was placed 2 months later followed by an immediate post-delivery panoramic radiograph (Figure 6g). In the sagittal and coronal images of CBCT taken 1 year after the prosthesis was delivered, the space between the bone bridge and the lower bone graft substitute was well ossified, and there was no surface bone resorption in the bone bridge (Figure 6h,i). In preoperative CBCT coronal images, a moderately thick lateral wall was maintained, and PSAA was not observed. No mucosal thickening of the maxillary sinus was observed. (Figure 6j). A slight thickening of the sinus mucosa was observed in the CBCT coronal image taken after obtaining the lateral window (Figure 6k). In the CBCT coronal image taken 1 year after prosthesis delivery, the size of the bony window decreased, and no changes were observed in the maxillary sinus (Figure 6l).

### 2.2. Case 2

This patient, a 48-year-old male smoker, developed severe bone resorption and tooth mobility due to advanced periodontitis in the right mandibular second molar (Figure 7a). After 2 months of tooth extraction, VRA with the bone bridge technique was performed. A simultaneous implant placement was also planned for the #26 tooth missing area. On the panoramic radiograph taken 2 months after tooth extraction, severe vertical ridge deficiency was evident (Figure 7b). A decision was made to use the lateral sinus bony window on the contralateral side as a source of bone block graft, as the implant placement at the #26 site was also planned.

The immobilization of the bone bridge was achieved by fixing the harvested bone to the proximal bone of the adjacent teeth by the press-fit method. Unlike the previous case, the gap between the defect and the bone bridge was not filled with any bone graft substitute; the defect morphology was well circumscribed and contained. (Figure 7c). The panoramic radiograph taken 6 months after the bone bridge technique did not show any signs of surface resorption of the bone bridge (Figure 7d). The bone bridge was well incorporated with the adjacent native bone, and no mobility of the bone bridge was observed during the implant placement (Figure 7e). Uncovering was performed 4 months after implant placement, and prosthesis was delivered 2 months later. In the periapical radiograph taken 1 year after prosthesis delivery, crestal bone loss around the implant was not observed, and bone remodeling was well achieved in the space between the bone bridge and the post-extraction defect (Figure 7f). The healing of the donor site was observed on the coronal images of CBCT. In the coronal image of the CBCT taken after bone harvest, the lateral sinus window was very thick, and it was considered an ideal thickness (Figure 7g). No abnormal finding was observed in the maxillary sinus in the CBCT taken after the #26 implant prosthesis was delivered (Figure 7h). The donor site had fully healed with new bone as evident in the CBCT taken 1 year after the #26 implant prosthesis was delivered (Figure 7i).

### 2.3. Case 3

A 61-year-old male non-smoker without significant medical history had #16 extracted 3 months prior and presented a post-extraction socket with a severe vertical ridge deficiency (Figure 8a). The #17 tooth was also extracted, and the bone bridge technique was performed using the lateral bony window of the maxillary sinus along with particulate grafts to prevent any dead space formation (Figure 8b). After 6 months of healing, two Ø 4.3 × 10 mm SLA-textured implants (Implantium, Dentium, Suwon, Republic of Korea) were placed. The bone bridge was well integrated with the adjacent native bone. Uncovering was performed after 4 months, and the prosthesis was delivered after 2 months (Figure 8c). In the panoramic radiograph taken 7 years after prosthesis delivery, the crestal bone level was still well maintained, and the implants were stable (Figure 8d). No abnormal findings were found in follow-up CBCT images (Figure 8e,f). In the sagittal image of CBCT taken 7 years after the prosthesis was delivered, the crestal bone level around the implant was well maintained (Figure 8g). The donor site in the lateral wall of the maxillary sinus was healing with new bone formation to close the communication (Figure 8h–j).

### 2.4. Case 4

A healthy 42-year-old male non-smoker patient presented himself to the clinic for re-implantation of a #16 implant. The implant #16, which was placed 10 years ago, developed peri-implantitis with severe bone resorption and suppuration (Figure 9a). The failing implant was removed, and another implant was placed concurrently with additional transcrestal sinus floor elevation. No perforation of the sinus membrane occurred during the sinus floor elevation (Figure 9b). The removed bone from the window was used as a bone bridge for the vertical ridge augmentation of the explantation site (Figure 9c). The space between the bone bridge and the explantation defect was filled with particulate bone (Osteon III, Genoss, Suwon, Republic of Korea). The bone bridge was fixed to the adjacent native bone using the press-fit method (Figure 9d) and covered with a resorbable collagen membrane (Figure 9e). The mucoperiosteal flap was closed.

During the healing period, there were no adverse events at the bone graft site, but the wound was partially opened, and healing was delayed (Figure 9f). After 5 months, the mucoperiosteal flaps were reflected for implant placement. The bone bridge seemed to have resorbed slightly and soft tissue ingrowth was observed. However, the existing bone bridge was well integrated with the surrounding native bone and did not show any mobility. An osteotomy was prepared for a Ø 4.3 × 10 mm SLA-textures implant (Implantium, Dentium, Suwon, Republic of Korea), and the fixture was placed with good initial stability. The implant was positioned 1.0 mm subcrestally to the platform, and the healing abutment was inserted. (Figure 9g). The flap was closed with 4-0 nylon. Uncovering was performed after 4 months, and the healing process was uneventful. After 2 months, the final prosthesis was delivered. One year after the prosthesis was delivered, no resorption was observed in the crestal bone level, and chewing function was restored (Figure 9h) (Table 1).

## 3. Discussion

Using the bone bridge technique with bone from the lateral wall of the sinus presented adequate bone filling of single bone defects after tooth extraction in four clinical cases at the maxilla and mandible. Minimal complications were observed, and the gradual closure of the bone window was observed after 7 years in follow-up CBCT evaluations.

All the present cases show that bone defects after extractions still persisted despite adequate maturation of surrounding soft tissues. The cases also demonstrate that the bone bridge technique using bone blocks obtained from an ipsilateral/contralateral sinus bony window enabled the successful VRA of a compromised post-extraction socket with a localized vertical ridge deficiency. The lateral bony window was a suitable donor site with no notable complications, and the area healed well with new bone formation.

Although still challenging, there have been many notable advances in the application of techniques for VRA. For VRA, bone blocks are mainly used rather than particulate bone, and autogenous bone is preferred over allograft, xenograft, and synthetic graft [8]. In addition, the use of a barrier membrane is essential, and a polytetrafluoroethylene (PTFE) membrane is widely used over a resorbable membrane [14]. Nevertheless, infection due to early exposure of the non-absorbable membrane may adversely affect the postoperative outcome [15]. A resorbable collagen membrane may possess insufficient space-making capacity as compared to a non-absorbable PTFE membrane [16]. In order to reinforce the mechanical stability of the resorbable collagen membrane, pins, tenting screws, and titanium mesh can be additionally used, but a larger surgical field must be created, and the procedure is also technically sensitive [17]. Cross-linking the collagen membrane increases mechanical stability, extends the absorption time, and induces osteogenic bone formation [18,19,20]. In the present cases, the bone bridge also serves to maximize the mechanical stability of the cross-linked collagen membrane.

Classical techniques for VBA include onlay graft using autogenous bone blocks, distraction osteogenesis, and the use of particulate bones and PTFE membranes [21]. The bone ring technique and sausage technique have also been introduced [11,22]. Symphysis or ascending ramus as main donor sites have been widely used for ridge reconstruction for implant placement for a long time [11,23]. One of the challenges of this is that instrumental access is limited, and the patient’s morbidity is high. In addition, many postoperative complications may require donor site restoration [7,8,9]. Therefore, an alternative donor site was needed, and Sakkas et al. and Kuster et al. introduced the zygomatic alveolar crest [24,25]. It was reported that the zygomatic alveolar crest provides easy instrument access and less patient morbidity [24,25]. In comparison, the position of the lateral sinus window used in the bone bridge technique is relatively low, so the instrument is easier to access, and the patient’s morbidity is minimal. Recently, Park et al. reported the successful bone regeneration of post-extraction sockets using a lateral sinus window [13] and suggested that the bone from the lateral sinus window can be used for VRA.

For the bone bridge technique to be successful, the bone block obtained from the lateral wall of the sinus should be of sufficient size to enclose the bony defect and of thickness of more than 1.0mm so that graft immobilization can be achieved with the press-fit effect. Moreover, it is appropriate to evaluate the thickness of the lateral wall prior to the surgery to ensure proper thickness with a CBCT. Bone grafts obtained from various sites such as tori, ascending ramus, and symphysis can be used as donor sites for bone bridges. However, the lateral wall of the maxillary sinus seems to provide bone blocks that maintain a uniform thickness and shape.

The space between the bone bridge and the post-extraction socket is filled with particulate bone, which plays a role in minimizing the dead space and supporting the bone bridge [26]. Micro-CT and histopathological findings confirmed that the integration between the bone bridge and the particulate bone was well noted, and the formation of new bone around the bone graft particles was evident. In CBCT and micro-CT images of case 1, there was little surface resorption and volumetric change in the bone. This shows that the bone bridge technique is very suitable for maintaining ridge contour as well as for ridge augmentation. As new bone sprouts from the bone bridge progresses downward, osteogenesis concentrates in the vicinity part of the bone bridge. This is very similar to the bone regeneration process that occurs when repositioning the lateral sinus bony window during maxillary sinus augmentation [27].

Potential limitations may include the availability of bone of sufficient thickness from the lateral wall and the presence of a posterior superior alveolar artery (PSAA). Occasionally, a PSAA may be present and bleeding complications may occur, although no life-threatening hemorrhage has been reported [28]. In the cases of this study, no complication of the donor site occurred. Perforation of the sinus mucosa can occur during graft harvesting as well, but it is less likely than the sinus perforation that occurs during sinus floor elevation. Since the sinus membrane exposed at the donor site was preserved, osteogenic potential in the sinus membrane could be expected [29]. In fact, after a few months, the lateral wall from which the graft was obtained gradually filled in with new bone as evident in the CBCT images. There were no changes in the thickness of the sinus membrane or sinus morphology at the donor site after each surgery other than the transient thickening of the sinus membrane that was due to a normal inflammatory response.

For the primary closure of the flap, either the flap extension technique or periosteal releasing incision should be performed as is appropriate [30,31]. The healing period of the soft tissue of the post-extraction socket also requires at least 2 months. If the blood supply of the flap is insufficient or if the wound is exposed because of tension, successful augmentation becomes difficult because the bone bridge becomes sequestrated with delayed healing. The width and thickness of the keratinized mucosa are important, and the depth of the vestibular fornix should also be considered. In Case 4, there was delayed healing due to wound exposure and partial resorption of the bone bridge. Early exposure of wounds adversely affects clinical and radiological treatment outcomes and may require additional procedures.

In general, the implant procedure was performed at least 4–6 months after the autogenous bone block graft [8]. Implant placement was performed after 6 months of the bone graft having been performed in the presented cases. The surgical site had good bone formation and the bone bridge was well integrated with the adjacent native bone. The initial fixation at the time of the implant placement was very good. Standardized procedures for VRA are difficult to regularize, and various results may occur depending on the preference and experience of the operator. Currently, the gold standard procedure for VRA has not yet been determined [32].

For the cases presented in the manuscript, the bone bridge technique using the lateral sinus bony window was effective for the vertical ridge augmentation of the localized post-extraction socket. However, this is only an introduction to an alternative technique with limited case numbers. Additional case presentations and randomized studies should be continued in the future.

## 4. Conclusions

The bone bridge technique using the bone from the lateral sinus bony window can be applied to perform vertical bone regeneration from residual bone defects after single tooth extraction.

## Figures and Tables

**Figure 1 medicina-59-01626-f001:**
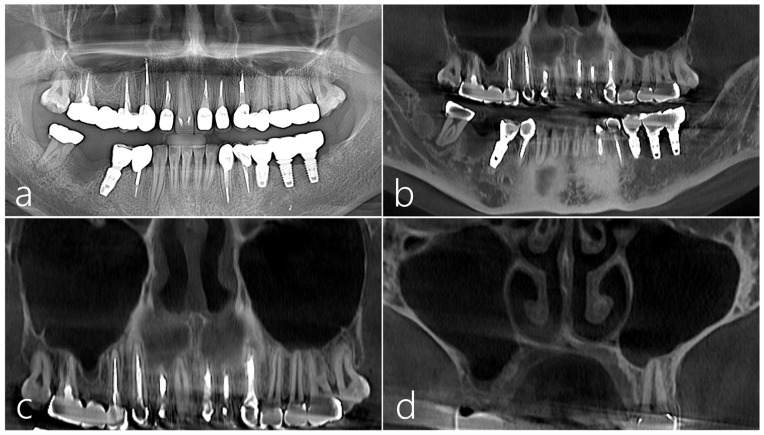
Case 1 (**a**) On the panoramic radiography taken before surgery, a severe vertical bone defect was observed in the extraction socket of the mandibular right first molar. Tooth extraction was performed 3 months prior due to periodontal disease; (**b**) A vertical ridge deficiency of the extraction socket was observed on a panoramic image of CBCT taken before surgery; (**c**,**d**) On the panoramic and coronal images of the CBCT taken before surgery, no lesion was observed in the right maxillary sinus, and the lateral sinus wall was maintained at an appropriate thickness. There was no PSAA.

**Figure 2 medicina-59-01626-f002:**
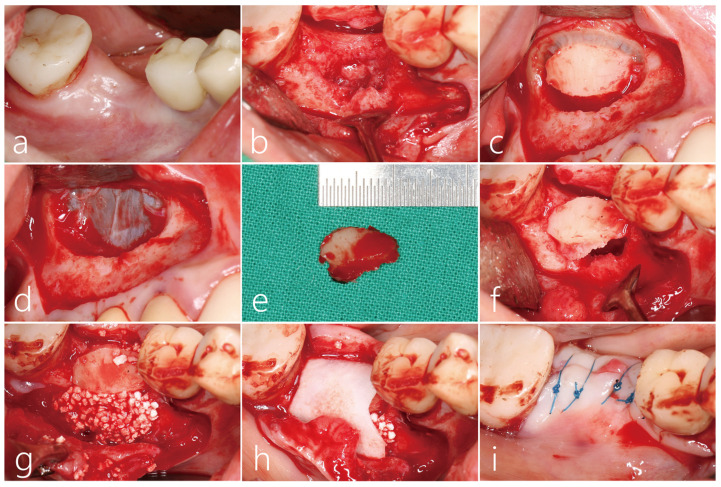
(**a**) Vertical deficiency was observed in the clinical picture 3 months after #46 tooth extraction; (**b**) The mucoperiosteal flap was reflected. An unhealed bone defect was observed in the extraction socket, but the proximal bone level of the adjacent teeth was well maintained. The lateral sinus window of the ipsilateral side was selected as the donor site; (**c**) A lateral sinus window larger than the size of the bone defect at site #46 was prepared; (**d**) The lateral sinus window was removed without perforation of the sinus membrane. The buccal flap was closed without membrane coverage; (**e**) The harvested lateral bone window was 2.0 × 1.5 mm in size. The sharp edge of the graft was trimmed; (**f**) The lateral bone window was fixed to the extraction socket site by the press-fit method. A gap occurred between the lateral bone window and the extraction socket; (**g**) The gap space was filled with a particulate type bone graft substitute; (**h**) The surgical site was covered with resorbable collagen membrane; (**i**) The flap was closed with 4-0 nylon.

**Figure 3 medicina-59-01626-f003:**
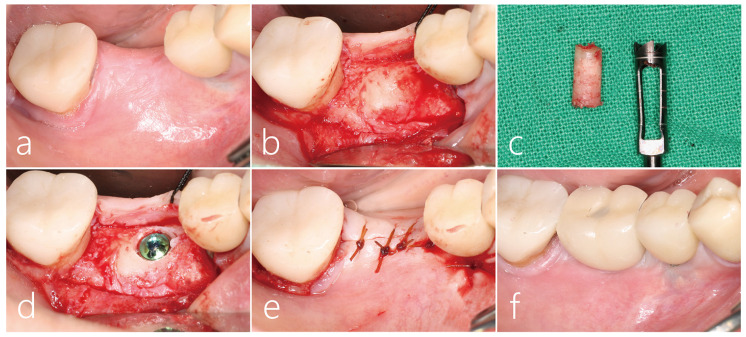
(**a**) Clinical findings after 6 months of bone graft showed that the collapsed soft tissue was remarkably vertically restored; (**b**) Under local anesthesia, the mucoperiosteal flap was reflected. The residual collagen membrane was removed. The bone bridge was well integrated with the adjacent native bone; (**c**) A core biopsy was performed with a trephine drill with a diameter of 3.0 mm at the point where the implant was to be placed. The bone bridge and the lower particulate bone were well fused with each other, and the dense bone quality was confirmed; (**d**) After an additional osteotomy, a Ø 4.3 × 10 mm SLA-textured implant was placed subcrestally at 1.0 mm; (**e**) The flap was closed with a 4-0 catgut; (**f**) Uncovering was performed 3 months after the implant placement, and the prosthesis was delivered 2 months later. Vertical bone deficiency was improved.

**Figure 4 medicina-59-01626-f004:**
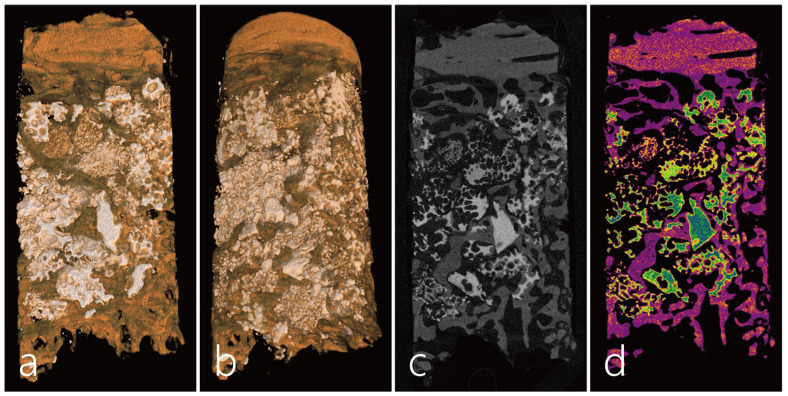
micro-CT images. (**a**,**b**) In the micro-CT image of the longitudinally scanned specimen, the upper portion is the bone bridge from the sinus (orange), middle portion is the residual particulate bone grafted site (white), new trabecular bone (brown), and the lower portion is the native bone. The upper bone bridge maintained its original thickness with almost no surface resorption, and the formation of new bone was concentrated in areas close to the bone bridge and native bone. (**c**,**d**) The formation of new bone was observed between the bone graft particles (green). There was also a lot of bone formation in the lower part of the specimen because it was adjacent to the native bone (pink).

**Figure 5 medicina-59-01626-f005:**
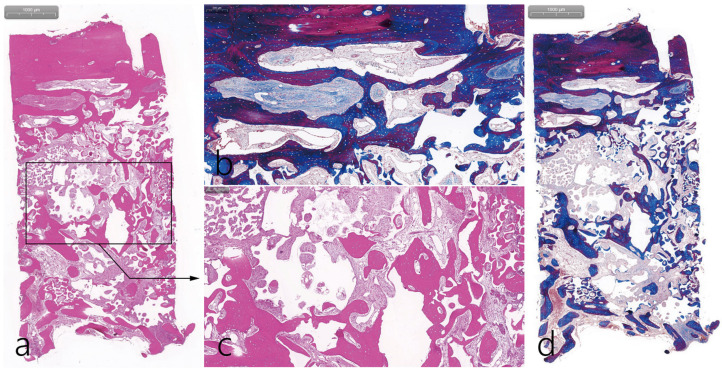
Histologic findings. (**a**) The upper part is the bone bridge, and the lower part is the post-extraction socket, which is native bone. The middle part is filled with particulate bone. It was found that bone formation started close to the bone bridge and native bone and moved to the particulate bone in the middle (H-E stain); (**b**) In a high-magnification view of the bone bridge, the new bone under the bone bridge was still immature bone. Capillaries, osteoblast, and osteocyte were observed in the new bone (MT stain); (**c**) Woven bone was observed around the particulate bone, and loose connective tissue was also observed (H-E stain); (**d**) New bone formation in contact with the bone bridge was observed (MT stain).

**Figure 6 medicina-59-01626-f006:**
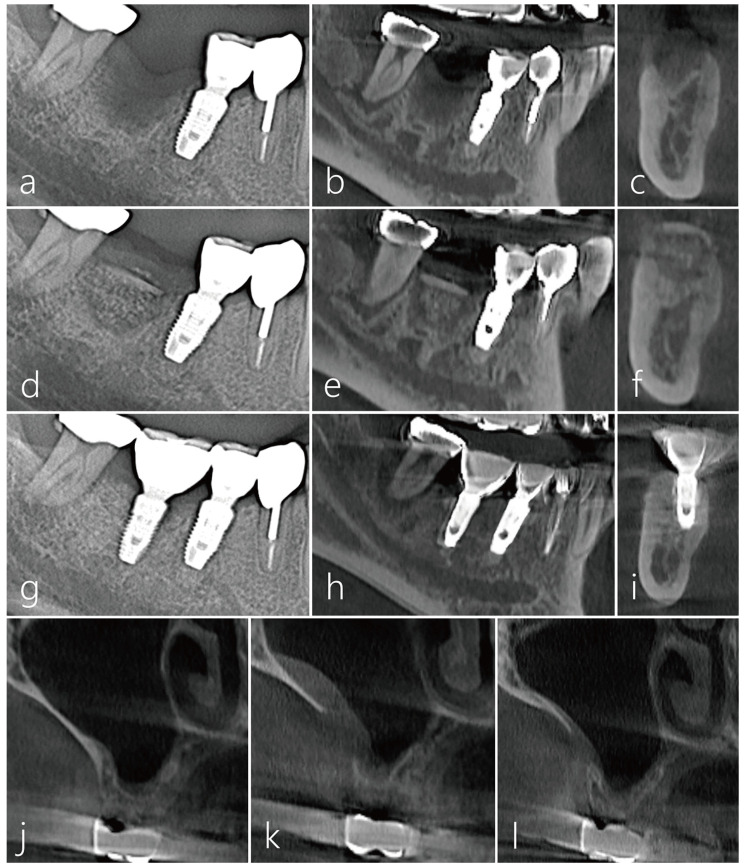
(**a**) In the panoramic radiography taken before surgery, the extraction socket of #46 tooth had a very severe bone defect; (**b**,**c**) A vertical ridge defect was observed in the sagittal and coronal images of the CBCT scanned from the extraction socket of #46 tooth; (**d**) Panoramic radiography taken after bone bridge procedure; (**e**,**f**) Bone bridge and particulate bone graft substitutes were observed in sagittal and coronal images of CBCT taken immediately after surgery; (**g**) The implant was placed 6 months after the bone bridge technique was performed. Uncovering was performed 3 months after implant placement, and the prosthesis was delivered 2 months later. Panoramic radiography taken after the prosthesis was delivered; (**h**,**i**) In the sagittal and coronal images of CBCT taken 1 year after the prosthesis was delivered, the space between the bone bridge and the lower bone graft substitute was well ossified, and there was no surface bone resorption in the bone bridge; (**j**) Coronal image of CBCT taken at the donor site preoperatively; (**k**) Coronal image of CBCT taken after the lateral window was removed. Slight thickening of the sinus mucosa was observed; (**l**) In the CBCT coronal image taken 1 year after prosthetic delivery, the size of the bony window decreased.

**Figure 7 medicina-59-01626-f007:**
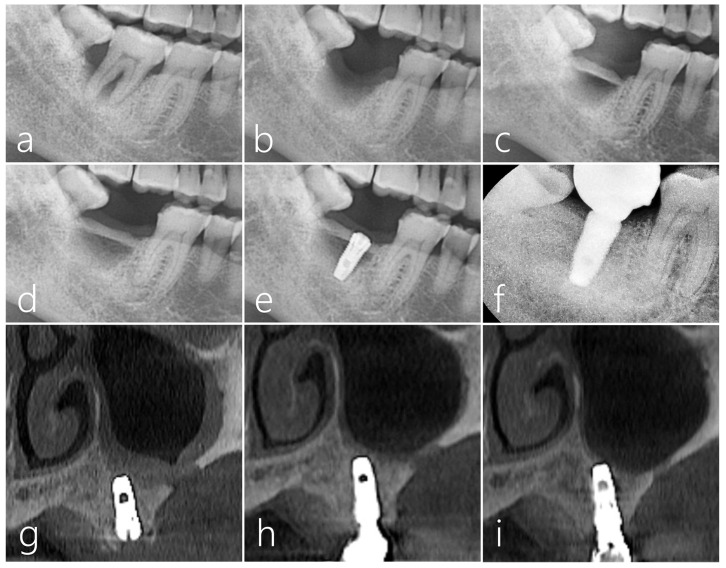
Case 2. (**a**) Severe bone resorption and tooth mobility occurred due to advanced periodontitis in #47 tooth; (**b**) On the panoramic radiography taken 2 months after tooth extraction, vertical ridge deficiency was very severe. A decision was made to use the lateral sinus bony window on the contralateral side as a bone block graft donor; (**c**) A bone bridge was achieved by immobilizing the harvested lateral sinus bony window to the proximal bone of the adjacent teeth by the press-fit method. The bone graft substitute was not filled in the gap space between the defect and the bone bridge; (**d**) On the panoramic radiography taken 6 months after the bone bridge technique, the transplanted bone was well incorporated with the adjacent native bone, and there was no movement during implant site preparation; (**e**) Panoramic radiography taken after implant placement; (**f**) In periapical radiography taken 1 year after prosthesis delivery, crestal bone loss around the implant was not observed, and bone remodeling was well done in the space between the bone bridge and the post-extraction socket; (**g**) CBCT image of the donor site taken immediately after surgery; (**h**) CBCT image of the donor site after prosthesis placement; (**i**) The size of the donor site was greatly reduced in the CBCT coronal image taken 1 year after the implant prosthesis was delivered.

**Figure 8 medicina-59-01626-f008:**
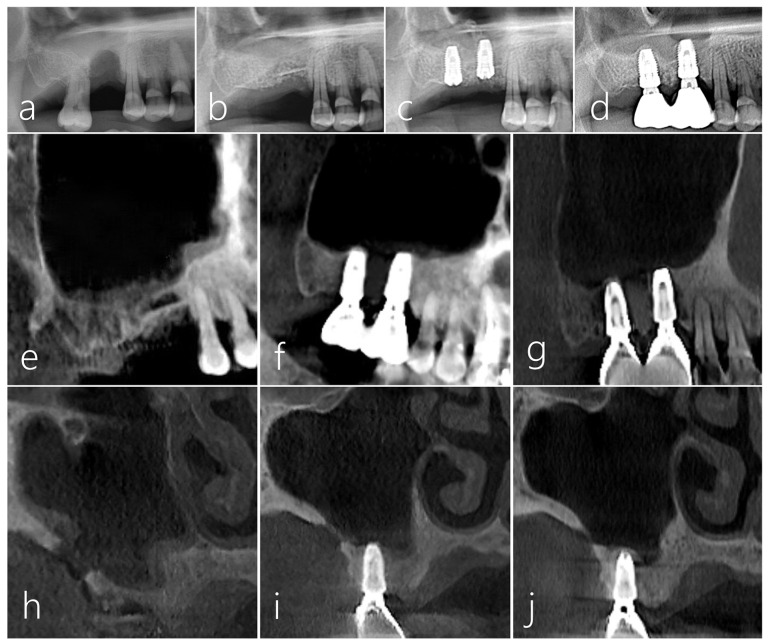
Case 3. (**a**) A localized post-extraction socket with severe vertical ridge deficiency was observed on a panoramic radiography taken 3 months after #16 tooth extraction; (**b**) The patient’s #17 tooth was also extracted, a bone bridge was prepared using the lateral sinus window of the maxillary sinus, and particulate bone was filled to reduce the dead space; (**c**) Two implants were placed 6 months after surgery. The bone bridge was well integrated with the adjacent native bone. Uncovering was performed after 4 months, and the prosthesis was delivered after 2 months; (**d**) Vertical bone augmentation of the compromised post-extraction socket was well achieved in the panoramic radiography taken 7 years after prosthesis delivery; (**e**) Sagittal image of CBCT taken immediately after the bone bridge technique; (**f**) Sagittal image of CBCT taken after prosthesis delivery; (**g**) In the CBCT sagittal image taken 7 years after prosthesis delivery, the crestal bone level around the implant was well maintained; (**h**) In the coronal image of CBCT taken after bone bridge technique, a donor site was observed; (**i**) The size of the donor site was reduced in the coronal image of CBCT taken after the prosthesis was delivered; (**j**) Only a small trace of the donor site was observed in the coronal image of CBCT taken 7 years after prosthesis delivery.

**Figure 9 medicina-59-01626-f009:**
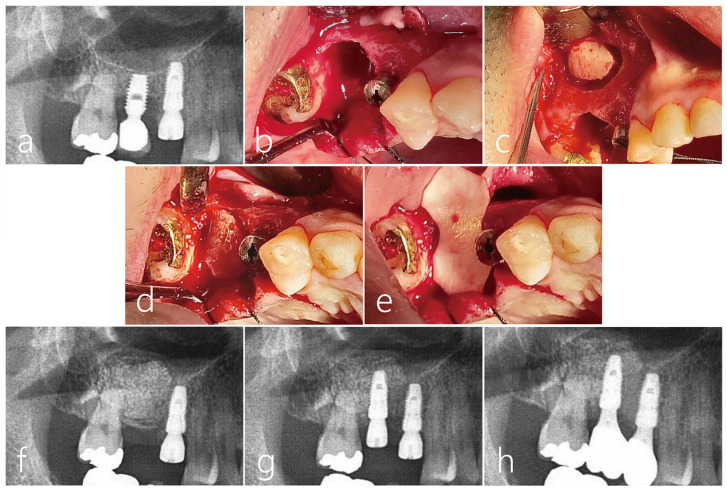
Case 4 (**a**) The patient #16 implant had peri-implantitis with severe bone resorption; (**b**) For a future successful implant replacement, the failed implant was removed and a transcrestal sinus floor elevation using Summers osteotome was performed simultaneously. (**c**) Sinus mucosa did not perforate during sinus floor elevation; (**d**) The removed lateral sinus access window was used as a bone bridge for vertical ridge augmentation of the explantation site. The dead space between the bone bridge and the explantation defect was filled with particulate bone. A bone graft was also performed in the elevated sinus. Bone bridges were fixed to the adjacent crestal bone by the press-fit method; (**e**) The graft site was covered with a resorbable collagen membrane. The flap was closed; (**f**) During the healing period, there was no special event at the bone graft site of the maxillary sinus, but the wound was partially opened, and healing was somewhat delayed; (**g**) After 5 months, an osteotomy was performed for implant site preparation. After thoroughly removing the infiltrated soft tissue, a Ø 4.3 × 10 mm SLA-textures implant was placed. At the time of the implant placement, the existing bone bridge was well integrated with the surrounding native bone and did not move. The implant was positioned 1.0 mm subcrestally to the platform. The flap was closed with 4-0 nylon. Uncovering was performed after 4 months, and the healing process was uneventful; (**h**) After 2 months, the final prosthesis was placed, and after 1 year, the crestal bone level was not resorbed, and chewing function was restored.

**Table 1 medicina-59-01626-t001:** Demographic information of the patients.

Case	Age/Sex	Smoking	Implant Sites	Implant (Diameter × Length)	Wound Exposure	Healing Period (Months)	Follow-Up Period
1	38/F	no	#46	4.3 × 10	no	6	1
2	48/M	yes	#47	4.3 × 10	no	6	1
3	61/M	no	#16 #17	4.3 × 10 4.3 × 10	no	6	7
4	42/M	no	#16	4.3 × 10	yes	5	1

## Data Availability

Not applicable.

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
