# Peer review of "The Bone Bridge Technique Utilizing Bone from the Lateral Wall of the Maxillary Sinus for Ridge Augmentation: Case Reports of a 1–7 Year Follow-Up"

_medicina, 2023, doi:10.3390/medicina59091626_

Round 1
Reviewer 1 Report
Dear Editor
Medicina, MDPI
Concerning the manuscript entitled “The Bone Bridge Technique Utilizing Bone from the Lateral 2 Wall of the Maxillary Sinus for Localized Vertical Ridge Aug-3 mentation in the Compromised Post-Extraction Socket. Case 4 Reports of 1-7 Year Follow-Up “, which I received for review.
The authors present an interesting clinical procedure that utilizes the cortical bone of the lateral wall of the maxillary sinus to graft vertical bone defects after teeth or implant extraction. The technique is innovative, and the presented clinical cases are the earliest reports with 1-7 years of follow-up known by this reviewer.
However, there are some clarifications and questions that require attention before publication.
Comments to the title: NONE
Comments to Case 1:
-The authors commented that a periapical and distal lesion was observed in tooth #47. Was that problem addressed? The lesion still is seen after the graft is inserted and the implant is restored. Are there any risks the authors envision by leaving this area near the graft and implant? Please clarify.
-Did the authors compress/pack the particulate graft for stability? The edges of the bone graft from the sinus were trimmed. They were also adjusted in form to fit the defect? In the clinical and radiographic images, the bone from the sinus was placed level with the adjacent bone peaks. Please provide details and clarify.
-The authors prepared the recipient area in some way. For example, removing soft tissue, irrigating, creating perforations, and curettage? Please detail and clarify.
-At reopening, why the authors removed the remaining membrane? Could it just be elevated and repositioned after the implant insertion? Please explain
-What was the stability of the implant at insertion? A cover screw was placed, right?
-In figures 4a to 4d.
This reviewer understands and is familiar with micro-ct images. But only some of the readers are. Please add landmarks (top, apical). Please explain the significance of the orange colour at the top (bone from the sinus), the white and green colour and (graft particles), the brown and pink colour (new trabecular bone) in figures a,b, and c and d.
-In figure 6k. Please describe briefly that a transient thickening of the sinus membrane adjacent to the window was observed. That returned to normal after some time (this an inflammatory response that is common in different sinus related procedures.
-Please explain why a membrane was not needed at the bony window created at the lateral wall of the sinus.
Comments to case 2:
In this case, the authors obtained the bone from the contralateral sinus. However, a justification needs to be included. For example, after the evaluation of a cone beam CT, the thickness was minimal, or the roots of the teeth were too close to the surface. This is important to understand the case.
In addition, please describe the specific donor area second premolar, first molar, second molar regions?
If the defect is not filled, no bone graft particles are inserted, and the four walls are present, the bone from the sinus is only used as an occlusive barrier. This is not vertical augmentation but regeneration. Also, please describe how the recipient area was treated before placing the bone cover. Did the authors induce bleeding from the area before inserting the bone cover? Was the membrane not placed?
This case is another example of well healed soft tissues essential to provide the primary closure. This seems to be a requirement for the technique.
No clinical pictures are presented from case 2. Please include the missing pictures.
Comments to case 3:
The authors present a radiographic evaluation. There are no clinical photos of case 3. This reviewer understands the narrative but, without the clinical pictures the characteristics of the defect, the soft tissues aspect and conditions and how the area was managed before the bone cover was placed is missing.
Comments to case 4:
None
Comments to results:
Please edit the following paragraph “These cases demonstrate that the bone bridging technique can result in successful VRA outcomes where an island of bone detached from the lateral sinus bony window is utilized to protect and maintain the space. Wound exposure occurred in 1 out of 4 cases, resulting in delayed healing and partial resorption of the bone bridge. Other than transient swelling and hematoma of the donor site, postoperative complications were mild. Long-term follow-ups also revealed that the donor sites healed with new bone as evident in the post-treatment CBCTs.”
And replace by,
“Using the bone bridge technique with bone from the lateral wall of the sinus presented adequate bone filling of single bone defects after tooth extraction in four clinical cases at the maxilla and mandible. Minimal complications were observed and gradual closure of the bone window was observed after 7 years in follo-up CBCT evaluations”.
Comments to discussion:
To clarify, in all the presented cases the soft tissues of the post-extraction areas were fully healed but, a bone defect remained. If so, that must be discussed and included in the conclusions.
For organization, this reviewer recommends to present clearly the indications and contraindications of this technique in a table. In addition, please add the requirements of the donor site and the defect area for this technique to be more successful.
Comments to conclusions:
Please edit and replace as follows
“The bone bridge technique using the bone from the lateral sinus bony window can be applied to perform vertical bone regeneration from residual bone defects after single tooth extraction”
Minor English editing is needed
Author Response
Thank you very much for all your comments and suggestions. Please see below for point-by-point responses to your questions.
Comments to Case 1:
-The authors commented that a periapical and distal lesion was observed in tooth #47. Was that problem addressed? The lesion still is seen after the graft is inserted and the implant is restored. Are there any risks the authors envision by leaving this area near the graft and implant? Please clarify.
Ideally, the lesion should have been addressed, but the patient refused to receive any treatment as the tooth was asymptomatic for a long time. As long as the graft and implant are not in close proximity to the lesion, we thought the procedure was safe.
-Did the authors compress/pack the particulate graft for stability? The edges of the bone graft from the sinus were trimmed. They were also adjusted in form to fit the defect? In the clinical and radiographic images, the bone from the sinus was placed level with the adjacent bone peaks. Please provide details and clarify.
Yes to all the questions. The bony window was trimmed properly for better adaptation in the recipient site. As the bone regeneration is expected not to go beyond the adjacent bone peaks, we placed the bone from the sinus at the same level.
-The authors prepared the recipient area in some way. For example, removing soft tissue, irrigating, creating perforations, and curettage? Please detail and clarify.
For the recipient site preparation, thorough debridement of granulomatous tissues from the socket was carried out followed by irrigation.
-At reopening, why the authors removed the remaining membrane? Could it just be elevated and repositioned after the implant insertion? Please explain
As the bone regeneration was complete, we believed that the membrane was no longer needed.
-What was the stability of the implant at insertion? A cover screw was placed, right?
The stability of the implant at insertion was very high as the bone was quite dense. Yes, a cover screw was placed.
-In figures 4a to 4d.
This reviewer understands and is familiar with micro-ct images. But only some of the readers are. Please add landmarks (top, apical). Please explain the significance of the orange colour at the top (bone from the sinus), the white and green colour and (graft particles), the brown and pink colour (new trabecular bone) in figures a,b, and c and d.
Corrected as suggested.
-In figure 6k. Please describe briefly that a transient thickening of the sinus membrane adjacent to the window was observed. That returned to normal after some time (this an inflammatory response that is common in different sinus related procedures.
Added to #404
-Please explain why a membrane was not needed at the bony window created at the lateral wall of the sinus.
Although a resorbable membrane is routinely used to cover the grafted window, a recent systematic review has shown that the amount of vital bone formation was not reduced even if a membrane was not used. This is also based on the clinical experiences of the authors.
Comments to case 2:
In this case, the authors obtained the bone from the contralateral sinus. However, a justification needs to be included. For example, after the evaluation of a cone beam CT, the thickness was minimal, or the roots of the teeth were too close to the surface. This is important to understand the case.
#223: It was decided to use the lateral sinus bony window on the contralateral side as a source of block bone graft as the implant placement at the #26 site was also planned.
In addition, please describe the specific donor area second premolar, first molar, second molar regions?
It was directly above the #26 site as described.
If the defect is not filled, no bone graft particles are inserted, and the four walls are present, the bone from the sinus is only used as an occlusive barrier. This is not vertical augmentation but regeneration. Also, please describe how the recipient area was treated before placing the bone cover. Did the authors induce bleeding from the area before inserting the bone cover? Was the membrane not placed?
#248: c) A bone bridge was achieved by immobilizing the harvested lateral sinus bony window to the proximal bone of the adjacent teeth by the press-fit method. The bone graft substitute was not filled in the gap space between the defect and the bone bridge
This case is another example of well healed soft tissues essential to provide the primary closure. This seems to be a requirement for the technique.
No clinical pictures are presented from case 2. Please include the missing pictures.
For Case #2, the authors believed that radiographs were sufficient to provide healing of the grafted site at different time points, and the quality of photographs was also not great.
Comments to case 3:
The authors present a radiographic evaluation. There are no clinical photos of case 3. This reviewer understands the narrative but, without the clinical pictures the characteristics of the defect, the soft tissues aspect and conditions and how the area was managed before the bone cover was placed is missing.
For Case #3, the authors believed that radiographs were sufficient to provide healing of the grafted site at different time points, and the quality of photographs was also not great.
Comments to case 4:
None
Comments to results:
Please edit the following paragraph “These cases demonstrate that the bone bridging technique can result in successful VRA outcomes where an island of bone detached from the lateral sinus bony window is utilized to protect and maintain the space. Wound exposure occurred in 1 out of 4 cases, resulting in delayed healing and partial resorption of the bone bridge. Other than transient swelling and hematoma of the donor site, postoperative complications were mild. Long-term follow-ups also revealed that the donor sites healed with new bone as evident in the post-treatment CBCTs.”
And replace by,
“Using the bone bridge technique with bone from the lateral wall of the sinus presented adequate bone filling of single bone defects after tooth extraction in four clinical cases at the maxilla and mandible. Minimal complications were observed and gradual closure of the bone window was observed after 7 years in follo-up CBCT evaluations”.
Corrected
Comments to discussion:
To clarify, in all the presented cases the soft tissues of the post-extraction areas were fully healed but, a bone defect remained. If so, that must be discussed and included in the conclusions.
Added to #339
For organization, this reviewer recommends to present clearly the indications and contraindications of this technique in a table. In addition, please add the requirements of the donor site and the defect area for this technique to be more successful.
As this is a case series with only 4 varying cases, future studies with larger sample sizes and more detailed analyses will be better suited to provide this information.
Comments to conclusions:
Please edit and replace as follows
“The bone bridge technique using the bone from the lateral sinus bony window can be applied to perform vertical bone regeneration from residual bone defects after single tooth extraction”
Replaced.
Reviewer 2 Report
The study introduces a novel technique for vertical ridge augmentation and presents clinical cases with long-term follow-up; its limitations in terms of sample size, lack of control group, and potential biases should be taken into consideration when interpreting the results. Further research with larger sample sizes and rigorous experimental designs would be needed better to understand the effectiveness and generalizability of the proposed technique.
Author Response
Thank you very much for your comments.
Reviewer 3 Report
Dear authors,
This case presentation presents interesting situations obtained through notable efforts. The paper contains sufficient data presented succinctly so as to guide the practitioner in similar approaches. However, I would like to make a few remarks, as follows:
Title
The title is very long, I suggest limiting it.
Abstract
The abstract is well structured and has sufficient data for orientation.
Introduction
The introduction contains sufficient data to frame the study and the technique used.
Matherial and methods
The material and method section should be introduced to specify the type of study, informed consent, and ethical approval, selection criteria, etc.
Clinical cases should be entered in the results section. The documented analysis was not for all cases and at 7 years, therefore it should be removed from the title. Do the authors have the patients' blood tests, possibly a vitamin D3 level?
Results
Presentations should be entered in the results section.
Discussion
The discussion section could be improved with the results of other studies, comparisons with the effectiveness of other techniques for similar time intervals.
References
No comment
Author Response
Title: The title is very long, I suggest limiting it.
Changed to “The Bone Bridge Technique Utilizing Bone from the Lateral Wall of the Maxillary Sinus for Ridge Augmentation. Case Reports of 1-7 Year Follow-Up”
Abstract
The abstract is well structured and has sufficient data for orientation.
Introduction
The introduction contains sufficient data to frame the study and the technique used.
Matherial and methods
The material and method section should be introduced to specify the type of study, informed consent, and ethical approval, selection criteria, etc.
As this is a case series, not a research study, each section contains all the pertinent information.
Clinical cases should be entered in the results section. The documented analysis was not for all cases and at 7 years, therefore it should be removed from the title. Do the authors have the patients' blood tests, possibly a vitamin D3 level?
The title indicates “1-7 year follow-up.” No, blood tests were not performed on these patients.
Results
Presentations should be entered in the results section.
For readers to follow each case more easily, the authors believed that each case presentation should include the introduction, procedure details, and the results, rather than separating them into the result section.
Discussion
The discussion section could be improved with the results of other studies, comparisons with the effectiveness of other techniques for similar time intervals.
This case report introduces a new technique, and there has not been any other study on this topic. The second and third paragraphs in the discussion section do provide some relevant literature.
References